# Detection of Mycoplasma Contamination in Transplanted Retinal Cells by Rapid and Sensitive Polymerase Chain Reaction Test

**DOI:** 10.3390/ijms222212555

**Published:** 2021-11-21

**Authors:** Sunao Sugita, Ayumi Hono, Shoko Fujino, Yoko Futatsugi, Yuta Yunomae, Norio Shimizu, Masayo Takahashi

**Affiliations:** 1RIKEN Center for Biosystems Dynamics Research, Laboratory for Retinal Regeneration, 2-2-3 Minatojima-minamimachi, Kobe 650-0047, Japan; ayumi.hono@riken.jp (A.H.); shoko.fujino@riken.jp (S.F.); yoko.futatsugi@riken.jp (Y.F.); retinalab@ml.riken.jp (M.T.); 2Vision Care Inc., Kobe 650-0047, Japan; 3Center for Stem Cell and Regenerative Medicine, Tokyo Medical and Dental University, Tokyo 113-8510, Japan; y-yunomae@nissui-pharm.jp (Y.Y.); nshivir@tmd.ac.jp (N.S.); 4Nissui Pharmaceutical Co., Ltd., Tokyo 110-8736, Japan

**Keywords:** mycoplasma, polymerase chain reaction, iPS cells, retinal cells, clinical trial

## Abstract

Contamination of cells/tissues by infectious pathogens (e.g., fungi, viruses, or bacteria, including mycoplasma) is a major problem in cell-based transplantation. In this study, we tested a polymerase chain reaction (PCR) method to provide rapid, simple, and sensitive detection of mycoplasma contamination in laboratory cultures for clinical use. This mycoplasma PCR system covers the *Mycoplasma* species (spp.) listed for testing in the 17th revision of the Japanese Pharmacopoeia, and we designed it for use in transplantable retinal cells. Here, we analyzed mycoplasma contamination in induced pluripotent stem cell (iPS cell)-derived transplantable retinal pigment epithelium (RPE) cells. In the spike tests to RPE cells with nine species of class *Mollicutes* bacteria, including seven *Mycoplasma* spp. and one of each *Acholeplasma* spp. and *Ureaplasma* spp., contamination at the concentration of 100 and 10 CFU/mL were detected with 100% probability in all cases, while 1 CFU/mL had a detection rate of 0–75%. DNA prepared from bacteria species other than class *Mollicutes* species was not detectable, indicating the specificity of this PCR. While iPS cells and iPS-RPE cells established in our laboratory were all negative by this PCR, some of the commercially available cell lines were positive. Cells for transplantation should never have infection, as once pathogens are implanted into the eyes, they can cause severe intraocular inflammation. Thus, it is imperative to monitor for infections in the transplants, although generally, mycoplasma infection is difficult to detect.

## 1. Introduction

Induced pluripotent stem cells (iPS cells) can differentiate into several types of cells or tissues, and they are now used in clinical studies/trials. In our laboratory, we have been investigating human iPS cell-derived retinal pigment epithelium (RPE) cells, and we successfully transplanted iPS-RPE cell sheets into age-related macular degeneration patients in 2014 [1]. In 2017, a clinical study was conducted with allogeneic transplants rather than autologous transplants due to cost and other considerations, and HLA homozygote iPS-RPE cells were transplanted into five HLA-matched patients with age-related macular degeneration [2]. Although these clinical trials were successful and did not accompany any obvious problems, cell transplantation required the utmost care to avoid contamination of the transplants by pathogens, as transplanting infected cells into the eye could cause severe adverse events in patients.

Generation of iPS cells and their differentiation into retinal cells, including RPE cells, require long-term culture (at least 8 months) with several passages, which demands consideration of possible contamination of cells with pathogenic microorganisms. Among them, the highest caution is required concerning infection by *Mycoplasma* species (spp.), as there is always a risk of transplanting mycoplasma-infected cells or tissues, which are hard to detect. In Japan, the 17th revision of the Japanese Pharmacopoeia defined the criteria for a mycoplasma contamination test. Conventional mycoplasma contamination tests, such as the culture method, DNA staining (by use of cultured Vero cells), and ELISA, have been used as standard tests. In addition, polymerase chain reaction (PCR) analysis (single-step, two-step nested PCR, or real-time PCR) is now being used. However, these examinations are time-consuming, especially for the culture method and the DNA staining method, and the procedures of PCR are often complicated [3,4,5,6,7,8,9,10,11,12,13]. Therefore, the purpose of this study was to examine whether rapid PCR could be applied to evaluate mycoplasma contamination in human iPS cell-derived transplantable RPE cells with the following criteria: (1) to serve as a comprehensive test of the genus *Mycoplasma* listed in the 17th Bureau Act, (2) to have high specificity and sensitivity, and (3) to be a rapid and simple procedure. In the present study, we applied a commercially available mycoplasma PCR system, the *Myco Finder* PCR, for validation on our transplantable RPE cells.

## 2. Results

### 2.1. PCR Sensitivity in the Mycoplasma PCR Test

First, we obtained the ratios of genomic copy (GC) number and colony-forming units (CFU) of the reference strains. For the assay, we used stocked strains of class *Mollicutes* including genus *Mycoplasma* (*Mycoplasma arginini*, *Mycoplasma fermentans*, *Mycoplasma hyorhinis**,*
*Mycoplasma salivarium**, Mycoplasma orale, Mycoplasma pneumoniae,* and *Mycoplasma hominis*), genus *Ureaplasma* (*Ureaplasma urealyticum*), and genus *Acholeplasma* (*Acholeplasma laidlawii*). As a result, all tested strains demonstrated low GC/CFU ratios (<100), which was less than 10 (GC/CFU ratios <10), except for *M. pneumoniae* (GC/CFU ratio = 30.59) (Table 1). Since the GC/CFU ratio is the ratio of the number of genomes against the number of live bacteria or fungal cells and the results for all strains were less than 100, this means for each strain, more than one live bacterium was present per 100 copies of the genome. When evaluating the sensitivity of mycoplasma tests based on nucleic acid amplification methods, 100 or less GC/CFU ratios of the reference product is desirable. Based on these results, we next examined the detection sensitivity of the *Myco Finder PCR* test.

For PCR sensitivity assays, the following strains were used: *A. laidlawii*, *M. hyorhinis*, *M. pneumoniae*, *M. orale*, *M. fermentans*, *M. salivarium*, *M.*
*arginini*, *M.*
*hominis*, and *U.*
*urealyticum*. Table 2 shows the sensitivities of the PCR for the nine *Mollicutes* spp. In the spike test in human iPS-RPE cells, 100 CFU/mL were detected with 100% probability in all cases, and 10 CFU/mL were also detected with 100% probability (Table 2). Surprisingly, spikes as low as 1 CFU/mL had a detection rate of 0–75%, which means that some *Mollicutes* spp. were detected even with a few enough numbers to form a single colony. Of note, the class *Mollicutes* spp., besides *Mycoplasma* spp., Acholeplasma spp. (*Acholeplasma laidlawii*; NBRC14400), and Ureaplasma spp. (*U. urealyticum*; ATCC 27618), was also detectable by this Myco Finder PCR assay (Table 2). In the spike test in iPS-RPE cells, 100 CFU/mL was detected with 100% probability.

Representative PCR results of the mycoplasma sensitivity test are shown in Figure 1. In iPS-RPE cells spiked with 10 CFU/mL of *M. arginini*, the mycoplasma was detected in 100% of the trials (four times out of four trials; Table 2), and the spike at the concentration of 1 CFU/mL was detected with a probability of 50% (two times out of four trials; Table 2). The PCR cycle required for the detection correlated with the concentration of the spiked mycoplasma (Figure 1), indicating the detection could be semi-quantitative.

### 2.2. Specificity of the Mycoplasma PCR Test

Next, the specificity of the mycoplasma PCR (*Myco Finder*) was verified by adding DNA from pathogens other than class *Mollicutes* species. For this assay, synthetic DNA of four bacteria other class *Mollicutes*, a fungus, and a virus were used. All six synthetic DNA templates, including those of bacterial species other than class *Mollicutes*, were not detected by this PCR (Figure 2A). Representative results of the specificity verification are presented in Figure 2B, which shows any of the examined DNA templates were undetectable in this PCR. To further confirm the specificity of the described mycoplasma PCR test, we used an additional 16 bacterial species [14], and all of them were not detected by this PCR (Table 3), confirming the high specificity of this method.

### 2.3. Representative Data of Mycoplasma-Positive Cells in Laboratory Cultures

To use this PCR test for laboratory management of contamination in cultured cells, we performed the test on many cell lines, including iPS-RPE cells. For each cell line, DNA was extracted and PCR was performed. The representative PCR data of mycoplasma-positive and -negative cells are shown in Figure 3A. The positive was a B cell line (B95-8).

When observed under an inverted microscope, the culture of the B95-8 B cell line that was mycoplasma-positive by PCR was indistinguishable from the one that was mycoplasma-negative by PCR (Figure 3B). By quantitative real-time PCR (qPCR), the infection of *Mycoplasma* spp. in these B95-8 cells was calculated as 4.4 × 10^6^ copies/mL. One of the cultures of commercially available human RPE cell line (ARPE-19) was also positive, but was apparently indistinguishable from uninfected batch by observation under amicroscope (Figure 3C). The copy number of *Mycoplasma* spp. calculated by qPCR in these infected ARPE-19 cells was low (9.3 × 10^2^ copies/μL) but still detectable. Albeit detected by PCR, these two examples of infected cells show it was difficult to determine mycoplasma contamination by microscopic observation.

With this PCR test, we confirmed that the iPS cells (n = 8) and iPS-RPE cells (n = 29) established in our laboratory were all negative. Primary RPE cells were also all negative (n = 4), but the ARPE-19 cell line was positive in one of the three cultures. B cell line (B95-8) was positive in one of the eight cultures. These results back up the notion that long-term cultured cells, such as cell lines, have increased risk of infection by class *Mollicutes*-bacteria, thus requiring extra attention.

### 2.4. Detection of Mycoplasma by SEM

We next examined the infected cells using scanning electron microscope (SEM) for the detection of mycoplasma. We harvested two cultured monkey iPS-RPE cells (line 46a) that were mycoplasma DNA-positive or -negative by *Myco Finder* PCR and observed them under SEM, which showed proliferating mycoplasma-like circular particles on infected RPE cells. While microvilli were observed on the cell surface of our established iPS-RPE cells under SEM (Figure 4A), we failed to find any microvilli on the cell surface of infected cells from six images of SEM (Figure 4B). In the magnified image of X30,000 (Figure 4B’), a large number of circular particles that seem to be mycoplasma with high electron density were observed. In contrast, no such particles were found in the control RPE cells that were mycoplasma-negative by *Myco Finder* PCR. From the entire view of one SEM image, we could see the infected cells were smaller overall compared to non-infected cells (Figure 4C). For evaluation of the size of infected and non-infected cells, as RPE cells were often larger than the field of view, instead of measuring the cell size, we counted the cell number in one field of view. The criteria for counting one cell were to have the nucleus in the field of view, without counting debris with obviously broken morphology (Figure 4D). As a result, the cell numbers of mycoplasma-positive cells were significantly higher than those of mycoplasma-negative cells, suggesting mycoplasma-infected cells had become smaller and lost RPE-specific cell morphology. By observing single units of mycoplasma under electron microscope, it was confirmed by PCR that mycoplasma-positive cells were indeed infected.

## 3. Discussion

We analyzed mycoplasma contamination in transplantable iPSC-derived RPE cells by rapid PCR examination. The results of PCR were negative in the iPS cells and iPS-RPE cells established in our laboratory but were positive in some of the cultures of commercially available cell lines. In the spike tests of nine bacterial species of class *Mollicutes* in RPE cells, 100 CFU/mL and 10 CFU/mL of spiked *Mollicutes* DNA were detected with 100% probability in all cases, and 1 CFU/mL had a detection rate of 0–75%. Additionally, bacterial DNA other than from class *Mollicutes* species was not detected by this PCR, indicating the test was not only sensitive but also highly specific.

Table 4 summarizes the comparison between the conventional tests for detecting mycoplasma contamination and *Myco Finder* PCR test. The conventional tests in Japan, which comply with the 16th revision of the Japanese Pharmacopoeia, are DNA staining and nested PCR. Real-time quantitative PCR is now also used [15]. These tests are often complicated and require experienced technicians. In addition, the sensitivity of DNA staining and nested PCR is 100 CFU/mL, and the average testing time for DNA staining is around 6 days. In contrast, *Myco Finder* PCR test, which complies with the 17th revision of the Japanese Pharmacopoeia, is semi-quantitative and not complicated. Importantly, its sensitivity is as high as 10 CFU/mL, and the testing time is around 140 min (Table 4). In fact, for the clinical study of RPE transplantation, conventional tests (DNA staining and nested PCR) were performed two weeks before the operation, and *Myco Finder* PCR test was additionally conducted on the day of transplantation [2]. This allowed for the monitoring of the latest condition of the cells immediately before the transplantation, according to which decisions to discontinue the operation could be made in case infection was detected.

Mycoplasma contamination has high prevalence, with more than 20% of cell cultures infected [5], suggesting mycoplasma potentially has higher risk of infection than other bacteria, fungi, and viruses. Among the class *Mollicutes* bacteria, *M. arginini, M. fermentans, M. hominis, M. hyorhinis, M. orale,* and *Acholeplasma laidlawii* account for the majority of the infections [6]. Since mycoplasma infections are usually unrecognizable or invisible by observation under a microscope during routine cell culture, they are risk factors for cell-based transplantation. The conventional methods currently conducted for detection are DNA staining, single-based PCR (PCR for one target gene), nested PCR, ELISA, qPCR, and fluorescent in situ hybridization (FISH) [7,8,9,10,11,12,13]. However, as these tests are time-consuming, it is not realistic to perform the tests every time we find suspicious cultures.

We herein report a rapid detection test for class *Mollicutes* bacterial species that can be easily performed in the laboratory. This PCR assay is based on the hybridization of an oligonucleotide probe that binds to a sequence within the 16S ribosomal RNA (i.e., 16S rDNA) of mycoplasma, which, together with the PCR primers, specifically amplifies the DNA sequence coding the mycoplasma rRNA gene. Theoretically, all *Mycoplasma* spp.—which includes more than 200 species—can be identified using these primers and probe. Hybridization to abundantly present ribosomal RNA provides a highly sensitive method that is readily applicable for mandatory quality-control tests in cell-based transplantation.

We have previously reported the incidence of mycoplasma infection in the eye of a monkey after subretinal transplantation of iPS-derived RPE cells [15]. Ophthalmological examination showed mass lesions and retinal vasculitis in the subretina, and pathological examination demonstrated severe infiltration of inflammatory cells at the transplantation site (RPE grafts) and hematoma in the retina and the subretina. By immunohistochemistry, infiltrations of diverse inflammatory cells, including T/B lymphocytes, neutrophils, and NK cells, were observed in the grafted area, the retina, and the choroids in the eye [15]. The acute and severe intraocular inflammation in this monkey after the transplantation warned us that the administration of mycoplasma-infected cells could be very dangerous. In fact, animal models with direct injection of mycoplasma into the eye exhibit similar ocular inflammation [16,17], and there are several reports of human uveitis with mycoplasma infections [18,19,20]. More importantly, while the ocular findings caused by the rejection after RPE cell transplantation with allografts [21,22,23,24,25] and those caused by mycoplasma infection are similar, especially in the early stages. However, the treatments for these two types of intraocular inflammation are completely different, making it necessary to have a sensitive method for mycoplasma detection at early stage of infection. Altogether, mycoplasma could cause inflammation when entered into the eye, and the need for regular and rapid mycoplasma testing of graft cells/tissues before and after transplantation into the retina was reaffirmed.

## 4. Materials and Methods

### 4.1. Cells and DNA Extraction

Human iPS cells, human RPE cell lines (ARPE-19), B cell lines (B95-8), and human Ff-I iPS cell-derived RPE cell lines were used. Additionally, human iPS cell-derived RPE cells were established, as described previously [26,27], from the iPS cell line that was used in a previous clinical study (line QHJI01s04) [2]. For *Myco Finder* PCR assay, 5 × 10^5^ cells of each cell line were harvested in sterile PBS, followed by DNA extraction with an automatic extraction machine (Qiagen, Valencia, CA, USA) and a DNA extraction kit (QIAamp UCP DNA Micro Kit: Qiagen).

For spike tests, 1.6 × 10^7^ RPE cells derived from human Ff-I iPS cells were prepared in PBS. Then, each mycoplasma strain diluted in PBS was added at the final concentration of 100, 10, and 1 CFU/mL, followed by heat treatment with proteinase K and DNA extraction by an automatic extraction machine and DNA extraction kit (Qiagen).

The cultured cell lines were also observed under an inverted microscope (IX71, OLYMPUS, Tokyo, Japan).

### 4.2. Mycoplasma Strains

Seven strains of *Mycoplasma* spp. listed in the 17th revision of the Japanese Pharmacopoeia were used: *M. hyorhinis, M. pneumoniae, M. orale, M. fermentans, M. salivarium, M.*
*hominis,* and *M.*
*arginini*. We also prepared two more species from different genera of class *Mollicutes*, *A. laidlawii* and *U.*
*urealyticum*. The strains were purchased from the American Type Culture Collection (ATCC) and the NITE Biological Resource Center (NBRC) (Table 1).

### 4.3. CFU and GC Measurements

Genomic copy (GC)/colony-forming units (CFU) ratios were measured by National Institute of Health Sciences (NIHS) -based methods (n = 3). For CFU measurements, mycoplasma stocks stored at −80 °C were thawed, and a 10-fold dilution series up to 10^−7^ was prepared in liquid medium (Hayflick Liquid: Merck Millipore, Tokyo, Japan). One of the *Mycoplasma* spp. *M. pneumoniae* was shredded by passing through a 27G needle syringe (Terumo, Tokyo, Japan) 10 times before dilution. Other *Mycoplasma* spp. were diluted immediately after thawed. Five μL of each mycoplasma diluted solution was added dropwise to Hayflick agar (Merck Millipore) at two locations and cultured at 37 °C in aerobic culture (AnaeroPack CO_2_) for *A. laidlawii*, *M. arginine*, *M. hominis*, *M. hyorhinis*, *M. pneumoniae,* or *U.*
*urealyticum* and in anaerobic culture (AnaeroPack Kenki) for *M. orale*, *M. fermentans*, and *M. salivarium*. Then, the number of mycoplasma colonies was counted under the microscope (inverted microscope: NIKON Eclipse TI-HUBC, Tokyo, Japan), and CFU/mL was calculated. For GC measurements, mycoplasma solution was centrifuged (14,000× *g*, 4 °C, 30 min), and DNA was extracted from the pellets using the PureLink Genomic DNA Mini Kit (Life Technologies). The DNA content measured by optical density at 260 nm (NanoDrop 1000, Thremo Scientific, Waltham, MA, USA) was used to calculate the genome copy number with the following formula [28]:Copies/mL = [DNA contents (µg/mL)] × 10^6^ × (0.978 × 10^9^) / [genome size]

The average of three independent measurements was used for CFU and GC measurements, respectively.

### 4.4. PCR

PCR was performed using the *Myco Finder* Kit (Nissui Pharmaceutical Co., Ltd., Tokyo, Japan) and LightCycler 480 II instrument (Roche, Basel, Switzerland) with the conditions shown in Figure 5. In the spike tests, each of the nine strains (refer to Table 1) was added to iPS-RPE cells at the concentrations of 1, 10, and 100 CFU/mL, respectively, for sensitivity evaluation. To confirm the data, quantitative real-time PCR (qPCR) was performed with a set of primers and probe targeting the 16S rDNA of mycoplasma (Nihon Techno Service Co., Ltd.; Tokyo, Japan), as described previously [15]. Prior to spike tests, the cell culture used for the test was verified as “mycoplasma negative” by the above-described qPCR targeting the 16S rDNA.

For specificity evaluation, *Myco Finder* PCR was tested on whole or synthetic DNA of 22 pathogens, including 20 bacteria other than class *Mollicutes*, a fungus, and a virus: Bacterial 16S rDNA (*Staphylococcus aureus* 16S region), *P. acnes* rDNA (*Propionibacterium acnes* 16S region), Chlamydia (*Chlamydia pneumoniae* 16S-23S spacer region), Syphilis (*Treponema pallidum* TP47 gene region), Fungal 28S rDNA (*Cryptococcus* 28S region), and HSV-1 (*Herpes simplex virus 1* UL27 gene region). These synthetic DNA products were purchased from Nihon Techno Service Co., Ltd. The result was similar to the specificity data provided by the vendor of *Myco Finder* PCR on their website (Nissui pharmaceuticals; https://www.nissui-pharm.co.jp/english/pdf/products/global/Pharmacopoeia_Validation.pdf). In addition, 16 bacteria (other than class *Mollicutes*) in Table 3 were tested on whole DNA, as described in our previous report [14].

### 4.5. Scanning Electron Microscope

For the detection of mycoplasma, we examined mycoplasma-infected retinal cells by scanning electron microscope (SEM). We harvested two cultured monkey iPS-RPE cells (line 46a) that were mycoplasma DNA-positive or -negative by *Myco Finder* PCR. 46a monkey iPS-RPE cells were previously established in our lab [21]. SEM observation was performed by Hanaichi UltraStructure Research Institute, Co., Ltd. (Aichi, Japan) with a HITACHI H-7600 (Hitachi High-Tech Co., Tokyo, Japan) electron microscope. For evaluation of the cell size, the cell number observable in one view was counted in 6 views of each culture. The criteria of counting one cell were to have the nucleus in the field, not counting debris with obviously broken morphology. The average of six views with standard deviation was used for quantification. For statistical analysis, *t*-test was used.

## 5. Conclusions

The Myco Finder Kit, a rapid and simple PCR system, had high sensitivity and specificity to detect class *Mollicutes* species such as M. hyorhinis, M. pneumoniae, M. orale, M. fermentans, M. salivarium, M. hominis, M. arginine, A. laidlawii, and U. urealyticum, which was applicable for the detection of mycoplasma infection in transplantable RPE cells. This method is worth evaluating in clinical trials/studies for the detection of mycoplasma in transplants.

## Figures and Tables

**Figure 1 ijms-22-12555-f001:**
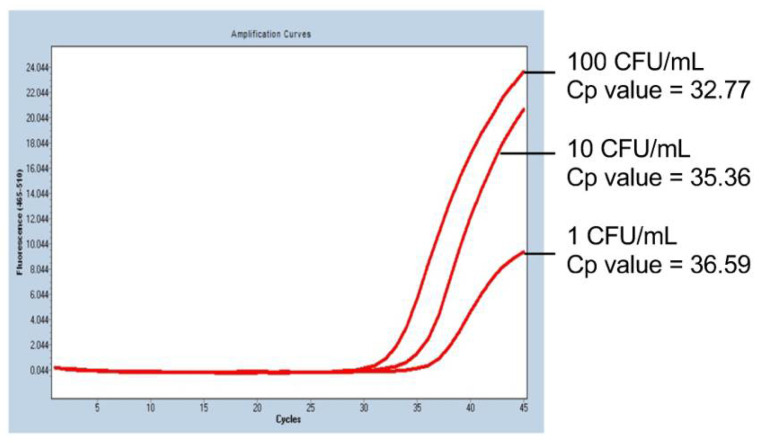
PCR results of the mycoplasma sensitivity test. In human iPS cell-derived RPE cells spiked with *M. arginini*, we observed that the number of PCR-cycles for the detection (red lines) depended on the concentration of the added *M. arginini*. Cp value: crossing point value.

**Figure 2 ijms-22-12555-f002:**
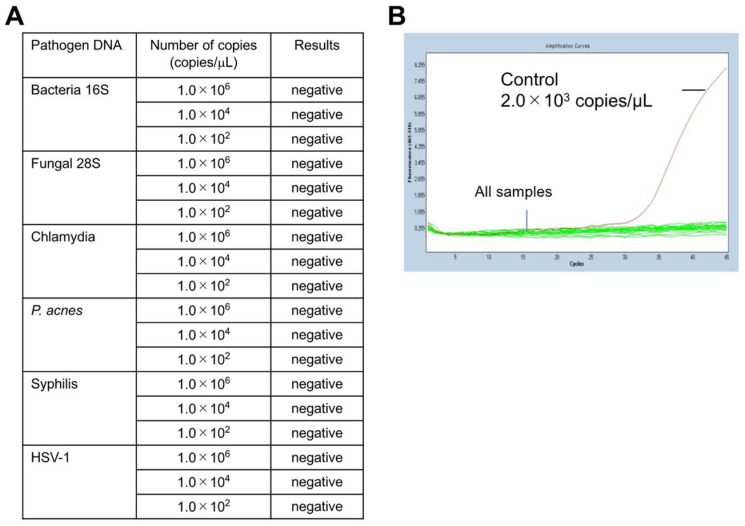
PCR results of mycoplasma specificity test. For this assay, synthetic DNA of bacterial species other than class *Mollicutes*, a fungus, and a virus were used: Bacterial 16S rDNA (*Staphylococcus aureus* 16S region), Fungal 28S rDNA (*Cryptococcus* 28S region), Chlamydia (*Chlamydia pneumoniae* 16S–23S spacer region), *P. acnes* rDNA (*Propionibacterium acnes* 16S region), Syphilis (*Treponema pallidum* TP47 gene region), and HSV-1 (*Herpes simplex virus 1* UL27 gene region). (**A**) All of the examined DNA templates were undetectable. (**B**) The detection level against the number of PCR-cycles shows all tested DNA, including bacterial DNA templates, was undetectable; the copy number of the positive control mycoplasma DNA (2 × 10^3^ copies/mL) was calculated by qPCR.

**Figure 3 ijms-22-12555-f003:**
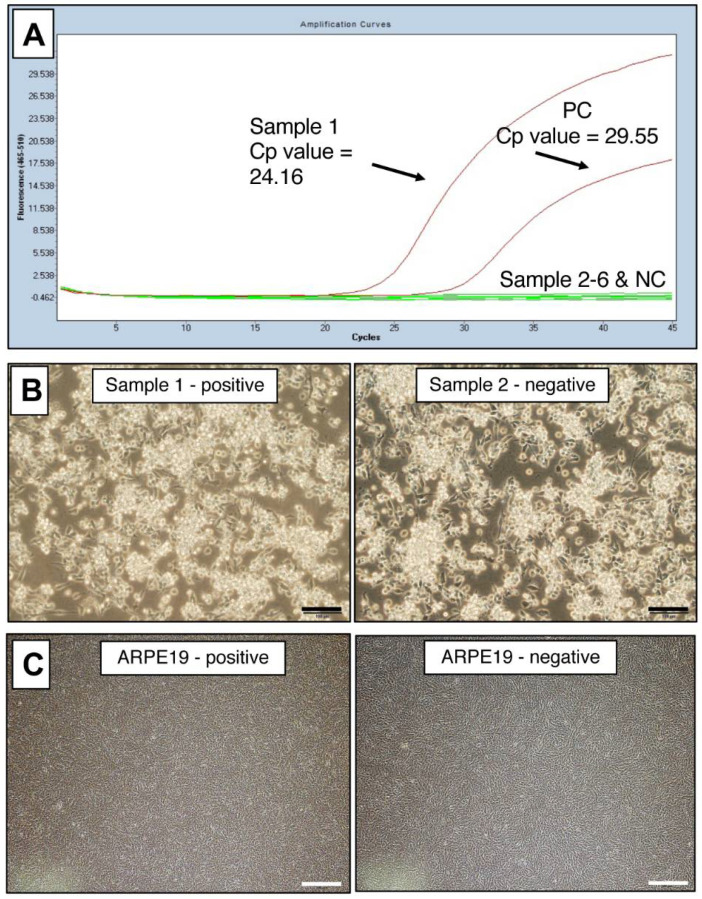
Representative PCR data of mycoplasma-positive cells. (**A**) *Myco Finder* PCR assay was performed on six cultures of three cell lines (B cell lines, iPS cells, and RPE cells) cultured in our laboratory. Sample 1 was positive, and Samples 2, 3, 4, 5, and 6 were negative. Cp value: crossing point value. (**B**) Positive and negative cultures of B95-8 B cell line were indistinguishable by observation under an inverted microscope: left (sample 1) was mycoplasma-positive, and right (Sample 2) was mycoplasma-negative by *Myco Finder* PCR. Scale bars: 100 μm. (**C**) Positive and negative cultures of a human RPE cell line (ARPE-19 cell line) were indistinguishable by the observation under an inverted microscope: left was mycoplasma-positive, and right was mycoplasma-negative by *Myco Finder* PCR. Scale bars: 500 μm.

**Figure 4 ijms-22-12555-f004:**
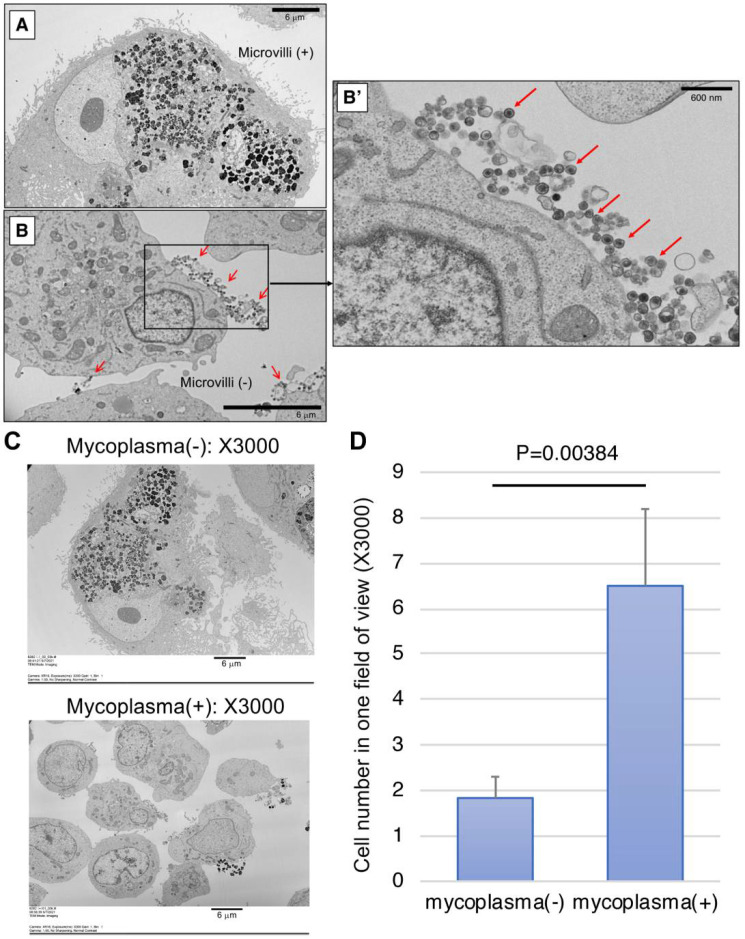
Representative pictures of scanning electron microscope (SEM) of mycoplasma-positive and -negative iPS-RPE cells. (**A**) Control iPS-RPE cells (mycoplasma-negative). Scale bar: 6 μm. (**B**) Mycoplasma-positive iPS-RPE cells. Scale bar: 6 μm. (**B’**) Magnified image (X30,000) of a mycoplasma-positive cell. Red arrows: high-density circular particles (=mycoplasma) detected under SEM. Scale bar: 600 nm. (**C**) The entire view of one SEM image (X3000). Upper: mycoplasma-positive RPE cells, lower: mycoplasma-negative RPE cells. (**D**) Cell numbers of mycoplasma-positive and -negative cells in one field of view (X3000). Cells with their nuclei in the view were counted, not counting debris with obviously broken morphology. The bar graph shows the average of six views with standard deviation *P* = 0.00384 between two groups (*t*-test; n = 6 for each group).

**Figure 5 ijms-22-12555-f005:**
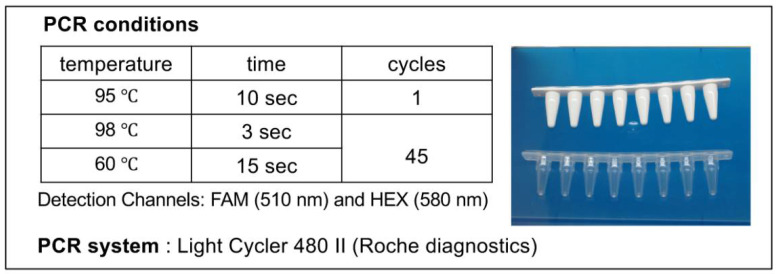
PCR Conditions for *Myco Finder* Kit. Samples were prepared in 8-well microtubes (**right**), and the DNA was amplified with LightCycler 480 II under the condition shown (**left**).

**Table 1 ijms-22-12555-t001:** Summary of class *Mollicutes* reference strains with ratios of GC and CFU.

*Mollicutes* Species	Strains	CFU/mL	GC/mL *	GC/CFU Ratios
*Mycoplasma arginini*	ATCC 23838	5 × 10^8^	1.36 × 10^9^	2.72
*Mycoplasma hyorhinis*	NBRC 14858	8.83 × 10^8^	1.9 × 10^9^	2.15
*Mycoplasma pneumoniae*	NBRC 14401	5.1 × 10^7^	1.56 × 10^9^	30.59
*Acholeplasma laidlawii*	NBRC 14400	1.07 × 10^8^	1.06 × 10^9^	9.91
*Mycoplasma fermentans*	NBRC 14854	1.04 × 10^9^	3.76 × 10^9^	3.62
*Mycoplasma salivarium*	NBRC 14478	1.41 × 10^9^	2.1 × 10^9^	1.49
*Mycoplasma orale*	NBRC 14477	6.37 × 10^8^	2.32 × 10^9^	3.64
*Mycoplasma hominis*	NBRC 14850	4 × 10^7^	1.76 × 10^9^	4.55
*Ureaplasma urealyticum*	ATCC 27618	2.03 × 10^8^	7.2 × 10^8^	3.54

* GC/mL: genomic copies (GC)/mL.

**Table 2 ijms-22-12555-t002:** Results of *Mollicutes* spp. PCR sensitivity test.

*A. laidlawii*		
Spike (CFU/mL)	Run	Positive rate (%)
100	4/4 *	100
10	4/4	100
1	1/4	25
*M. arginini*		
Spike (CFU/mL)	Run	Positive rate (%)
100	4/4	100
10	4/4	100
1	2/4	50
*M. hyorhinis*		
Spike (CFU/mL)	Run	Positive rate (%)
100	4/4	100
10	4/4	100
1	0/4	0
*M. pneumoniae*		
Spike (CFU/mL)	Run	Positive rate (%)
100	4/4	100
10	4/4	100
1	3/4	75
*M. orale*		
Spike (CFU/mL)	Run	Positive rate (%)
100	4/4	100
10	4/4	100
1	3/4	75
*M. fermentans*		
Spike (CFU/mL)	Run	Positive rate (%)
100	4/4	100
10	4/4	100
1	3/4	75
*M. salivarium*		
Spike (CFU/mL)	Run	Positive rate (%)
100	4/4	100
10	4/4	100
1	1/4	25
*M. hominis*		
Spike (CFU/mL)	Run	Positive rate (%)
100	4/4	100
10	4/4	100
1	4/4	100
*U. urealyticum*		
Spike (CFU/mL)	Run	Positive rate (%)
100	4/4	100
10	1/4	25
1	0/4	0

* 4/4 indicates that it was detected four times out of four PCR tests.

**Table 3 ijms-22-12555-t003:** Detection of main causative bacterial agents of infectious endophthalmitis by *Myco Finder* PCR.

Strains	Clone No.	Number of Copies (Copies/mL)	PCR Results
Gram-positive strains			
*Staphylococcus aureus*	NBRC12732	1.0 × 10^4^	negative
MRSA	JCM8702	1.0 × 10^4^	negative
*Staphylococcus epidermidis*	JCM2414	1.0 × 10^4^	negative
*Streptococcus pyogenes*	RIMD 3123004	1.0 × 10^4^	negative
*Streptococcus sanguinis*	JCM5708	1.0 × 10^4^	negative
*Streptococcus pneumoniae*	NBRC102642	1.0 × 10^4^	negative
*Enterococcus faecalis*	JCM20313	1.0 × 10^4^	negative
*Corynebacterium diphtheriae*	JCM1310	1.0 × 10^4^	negative
*Bacillus cereus*	JCM20266	1.0 × 10^4^	negative
*Clostridium perfringens*	JCM1290	1.0 × 10^4^	negative
*Propionibacterium acnes*	JCM6425	1.0 × 10^4^	negative
*Nocardia asteroides*	NBRC14403	1.0 × 10^4^	negative
Gram-negative strains			
*Escherichia coli*	JCM20135	1.0 × 10^4^	negative
*Klebsiella pneumoniae*	JCM1662	1.0 × 10^4^	negative
*Pseudomonas aeruginosa*	JCM6425	1.0 × 10^4^	negative
*Moraxella lacunata*	JCM20914	1.0 × 10^4^	negative

MRSA—Methicillin-resistant *Staphylococcus aureus*.

**Table 4 ijms-22-12555-t004:** Comparing *Myco Finder* PCR test with conventional methods for detecting mycoplasma.

	*Myco Finder* PCR	DNA Staining	Nested PCR	Real-Time PCR **
Method	Semi-quantitative PCR	Staining	Qualitative PCR	Quantitative PCR
Testing	Simplified	Experienced	Experienced	Experienced
Specimens	Cells (≤1 × 10^6^)	Culture supernatant	Culture supernatant	Cells (≤5 × 10^6^)
Sensitivity	10 CFU/mL	100 CFU/mL	100 CFU/mL	10 CFU/mL
Testing time *	140 min	6 days	6 h	4~5 h
JapanesePharmacopoeia	17th revision	16th revision	16th revision	17th revision

Conventional methods indicate our previous conventional tests for detecting *Mycoplasma* spp. * Testing time—time from specimen submission until determination of the result. ** Real-time PCR for detecting *Mycoplasma* spp.

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
