# Peer review of "Detection of Mycoplasma Contamination in Transplanted Retinal Cells by Rapid and Sensitive Polymerase Chain Reaction Test"

_ijms, 2021, doi:10.3390/ijms222212555_

Round 1

Reviewer 1 Report

The authors addressed all the reviewer's comments. However, the names of microorganisms  (in italics or not), e.g. M. hominis are not still unified throughout the text of manuscript. 

Author Response

Reviewer #1.

The authors addressed all the reviewer's comments. However, the names of microorganisms (in italics or not), e.g. M. hominis are not still unified throughout the text of manuscript. 

Response: As per your comments, we revised the names of microorganisms to italic throughout the manuscript.

Reviewer 2 Report

Dear authors,

The manuscript „Detection of mycoplasma contamination in transplanted retinal cells by rapid and sensitive polymerase chain reaction” is generally understandable and concise. The examination with the use of assay proposed in this paper are important for human cell-based transplantology. The assay in the manuscript was examined properly and is useful in the effective detection of Mycoplasma spp. and some related Mollicutes in transplantable cells. The introduction is written concise and properly. In the “Materials and methods” subchapter the description of proposed assay is sometimes not clear. The authors do not properly use the nomenclature of bacteria. Acholeplasma laidlawii and Ureplasma urealitycium are not mycoplasmas, they belong to Mollicutes class and are closely related to Mycoplasma spp. The description of test are also confusing. The reader don’t know if the test was developed in the lab or was it a commercial test validated for the use in the lab. Nevertheless the test is valuable and its use in the examination of cell-based transplantation will be profitable.

Broad comments:

I suggest to add the information: “test” in the end of the title.

Acholeplasma laidlawii and Ureplasma urealitycium are not mycoplasmas, they belong to Mollicutes class and are closely related to Mycoplasma spp. Authors should verify it e.g. in lines: 66, 248, in the title of Table 1 and 2.

Specific comments:

Line 19: Change “monitoring” into “detection”.

Line 21: Name those Mycoplasma spp. that were checked for the detection with described PCR test. Change “genus Mycoplasma” into “Mycoplasma spp.”.

Line 31: Change “observe” into “detect”.

Line 47: give more adequate time frame than “e.g. 10 months” – minimum or average.

Line 50: In my opinion it is not “a risk of mycoplasma infection in cultured cells” but a risk of transplantation the mycoplasma-infected cells or tissues.

Lines 53, 55, 56: “tests” repeated too many times.

Line 55: It is not clear from the sentence if PCR and culture are time consuming or only culture. Please specify it. We can assume that PCR is less time consuming than culture, but not than other PCR, even nested PCR – in my opinion it is small difference, not worth mentioning.

Line 61: “Myco Finder PCR” – authors should clearly specify if the test was developed by them or is it commercial, validated test that was adjusted to the use in the lab.

Line 68: It is not clear why authors put U. urealyticum in the separate sentence.

Lines 79-81, 86: Write names of bacteria with italic. After “Mycoplasma spp.” add “A. laidlawii and U. urealyticum”.

Table 2: Explain why A. laidlawi is written with bold?

Fig. 1, Table 2: A mistake in the name - M. arginine.

Line 104: Change “sources from bacteria” into for example “isolates from other than Mollicutes bacteria”. “Exogenous DNAs including…” is confusing, authors should change it into “This DNA isolates..” here and in Figure 2.

Line 107: “Even bacterial DNA templates were undetectable” is unnecessary, please generalize that any of the examined DNA templates were undetectable.

Line 110: Change “this” into “described”.

Lines 107-108: The mycoplasma DNA used as positive control do not indicate the high specificity of the test. The positive and negative controls indicates that the reaction was done properly. That is why authors should omit the inf. That positive control was positive and negative was negative, it is obvious. Delete this information here and in the description of the Table 3, Fig. 2 and 3.

Figure 2: The “Exogenous antigen DNA” is unclear. What does it mean? Is it DNA from different than Mollicutes species that were used for checking the specificity? It should be clearly written in the table and its description.

Line 113: Change “sources” into “isolates”.

Lines 114-116:  Did authors obtained a whole DNA isolated from bacteria, fungus and virus or a fragment of DNA, ex. 16S rDNA from S. aureus. If it was whole DNA it should be changed.

Table 3: There should be added information – causative agent of what?

Line 128: The information “actual mycoplasma positive and negative cells” are not clear. What it mean actual?

Lines 142-143: The last sentence may contain information about an increased risk of mycoplasmal or mollicutes infection.

Figure 3: Authors may put such information “Positive and negative cultures of B95-8 B cell line were indistinguishable by the observation under an inverted microscope: Left (sample 1)

was mycoplasma-positive, and right (sample 2) was mycoplasma-negative by PCR” once in the end with the information for both cell lines.

Table 4: The used PCR test should have the name.

I suggest to split the column “conventional tests” into two separate to separately describe DNA staining and nested PCR.

Line 185: Describe “this new PCR” with more specific language, for example described here Myco Finder PCR test.

Line 187: Authors claim that DNA staining and nested PCR are complicated and need experienced technics. In my opinion nested PCR is not so complicated. It should be verified.

Line 205: “single-based PCR” what author mean by single-based? PCR for one target gene? Please speficy.

Line 207: “Clearly detected” – what authors mean by clearly detected?

Line 208: “We herein report a new rapid detection method for mycoplasma” it is better to use test than method. It is not completely new method. Add the inf. About A. laidlawii and U. urealyticum.

Line 218-219: “examination showed” is repeated.

Line 247: It should be changed from “Mycoplasma strains from seven species” into seven strain of Mycoplasma spp.

Line 264: Specify what microscope.

Line 276: In my opinion better refer to Table 1.

Line 281: Change “six DNA sources” into DNA isolates from six… and put the names of bacteria, virus and fungus in a correct order, because now it is mixed.

Fig. . FAM and HEX are nor strictly detection wavelenghts but canals.

Line 291-292: In my opinion the information “according to the 17th revision of the Japanese Pharmacopoeia” is unnecessary in the conclusion. It is better to put the species examined with the test.

Author Response

Reviewer #2.

Dear authors,

The manuscript „Detection of mycoplasma contamination in transplanted retinal cells by rapid and sensitive polymerase chain reaction” is generally understandable and concise. The examination with the use of assay proposed in this paper are important for human cell-based transplantology. The assay in the manuscript was examined properly and is useful in the effective detection of Mycoplasma spp. and some related Mollicutes in transplantable cells. The introduction is written concise and properly. In the “Materials and methods” subchapter the description of proposed assay is sometimes not clear. The authors do not properly use the nomenclature of bacteria. Acholeplasma laidlawii and Ureplasma urealitycium are not mycoplasmas, they belong to Mollicutes class and are closely related to Mycoplasma spp. The description of test are also confusing. The reader don’t know if the test was developed in the lab or was it a commercial test validated for the use in the lab. Nevertheless the test is valuable and its use in the examination of cell-based transplantation will be profitable.

Response: Thank you for your comments. We really appreciate. We have revised the manuscript again as per your suggestions as follows.

Broad comments:

I suggest to add the information: “test” in the end of the title.

Response: As per your suggestion, we added “test” to the end of the title.

Acholeplasma laidlawii and Ureplasma urealitycium are not mycoplasmas, they belong to Mollicutes class and are closely related to Mycoplasma spp. Authors should verify it e.g. in lines: 66, 248, in the title of Table 1 and 2.

Response: We used the term mycoplasma to denote the species in class Mollicutes, but to be more accurate we clarified that we used class Mollicutes species including Acholeplasma laidlawii and Ureplasma urealitycium (lines 68-72; Table 1 and 2).

Specific comments:

Line 19: Change “monitoring” into “detection”.

Response: We changed it.

Line 21: Name those Mycoplasma spp. that were checked for the detection with described PCR test. Change “genus Mycoplasma” into “Mycoplasma spp.”.

Response: The Mycoplasma spp. tested in this study is listed in the main text. We do not think it is appropriate to list the names in the abstract. We changed “genus Mycoplasma” to “Mycoplasma spp.” as you suggested.

Line 31: Change “observe” into “detect”.

Response: We changed it.

Line 47: give more adequate time frame than “e.g. 10 months” – minimum or average.

Response: We changed it.

Line 50: In my opinion it is not “a risk of mycoplasma infection in cultured cells” but a risk of transplantation the mycoplasma-infected cells or tissues.

Response: We changed it as you suggested.

Lines 53, 55, 56: “tests” repeated too many times.

Response: We changed some of the “tests”.

Line 55: It is not clear from the sentence if PCR and culture are time consuming or only culture. Please specify it. We can assume that PCR is less time consuming than culture, but not than other PCR, even nested PCR – in my opinion it is small difference, not worth mentioning.

Response: We agree with your comments. Therefore, we revised the sentence.

Line 61: “Myco Finder PCR” – authors should clearly specify if the test was developed by them or is it commercial, validated test that was adjusted to the use in the lab.

Response: Myco Finder PCR test is the latter. For clarification, we added the following sentence: In the present study, we applied a commercially available mycoplasma PCR system, Myco Finder PCR, for validation on our transplantable RPE cells.

Line 68: It is not clear why authors put U. urealyticum in the separate sentence.

Response: We put it in the same sentence with other Mollicutes species.

Lines 79-81, 86: Write names of bacteria with italic. After “Mycoplasma spp.” add “A. laidlawii and U. urealyticum”.

Response: We revised the names of microorganisms to be written in italic throughout the manuscript.

Table 2: Explain why A. laidlawi is written with bold?

Response: That was just an error. We revised it.

Fig. 1, Table 2: A mistake in the name - M. arginine.

Response: That was a typo. We revised it through the manuscript.

Line 104: Change “sources from bacteria” into for example “isolates from other than Mollicutes-bacteria”.  “Exogenous DNAs including…” is confusing, authors should change it into “This DNA isolates..” here and in Figure 2.

Response: As your suggestions, we have changed the term “exogenous DNA” to “synthetic DNA”, including Fig. 2 Legend and the method section.

Line 107: “Even bacterial DNA templates were undetectable” is unnecessary, please generalize that any of the examined DNA templates were undetectable.

Response: We have changed the sentences including Fig. 2 Legend.

Line 110: Change “this” into “described”.

Response: We have changed the word.

Lines 107-108: The mycoplasma DNA used as positive control do not indicate the high specificity of the test. The positive and negative controls indicates that the reaction was done properly. That is why authors should omit the inf. That positive control was positive and negative was negative, it is obvious. Delete this information here and in the description of the Table 3, Fig. 2 and 3.

Response: As per your suggestions, we omitted the information regarding the positive and negative controls in the revised manuscript (text, Table 3, Fig. 2, and Fig. 3).

Figure 2: The “Exogenous antigen DNA” is unclear. What does it mean? Is it DNA from different than Mollicutes species that were used for checking the specificity? It should be clearly written in the table and its description.

Response: We are sorry for the confusion. What we meant by “exogenous DNA” was synthetic DNA used for this experiment. We clarified this in the revised manuscript (text, Fig. 2 legend, methods). In new Fig. 2, we changed “Exogenous antigen DNA” to “Pathogen DNA”.

Line 113: Change “sources” into “isolates”.

Response: We changed “DNA sources” to “synthetic DNA”.

Lines 114-116:  Did authors obtained a whole DNA isolated from bacteria, fungus and virus or a fragment of DNA, ex. 16S rDNA from S. aureus. If it was whole DNA it should be changed.

Response: These were not whole DNA but synthetic DNA products purchased from Nihon Techno Service Co., Ltd. (Tokyo, Japan). We added the information in the method section 4.4.

Table 3: There should be added information – causative agent of what?

Response: These were causative agents of “infectious endophthalmitis” (=bacteria-related infectious intraocular inflammatory disorders). We included the information in the title of Table 3.

Line 128: The information “actual mycoplasma positive and negative cells” are not clear. What it mean actual?

Response: We are sorry for the confusion. We omitted the word “actual” and revised the sentence: The representative PCR data of mycoplasma-positive and -negative cells are shown in Fig. 3A.

Lines 142-143: The last sentence may contain information about an increased risk of mycoplasmal or mollicutes infection.

Response: We agree with your comments. We changed the last sentence as you suggested.

Figure 3: Authors may put such information “Positive and negative cultures of B95-8 B cell line were indistinguishable by the observation under an inverted microscope: Left (sample 1) was mycoplasma-positive, and right (sample 2) was mycoplasma-negative by PCR” once in the end with the information for both cell lines.

Response: The description “Positive and negative cultures of B95-8 B cell line were indistinguishable by the observation under an inverted microscope” was in the second paragraph of section 2.3, and “Left (sample 1) was mycoplasma-positive, and right (sample 2) was mycoplasma-negative by PCR” overlaps with the legend of Fig. 2. However, as per your suggestion, we modified the sentence in the second paragraph of section 2.3.

Table 4: The used PCR test should have the name. I suggest to split the column “conventional tests” into two separate to separately describe DNA staining and nested PCR.

Response: We revised Table 4 as per your suggestions.

Line 185: Describe “this new PCR” with more specific language, for example described here Myco Finder PCR test.

Response: We revised the words.

Line 187: Authors claim that DNA staining and nested PCR are complicated and need experienced technics. In my opinion nested PCR is not so complicated. It should be verified.

Response: We have revised the sentences and Table 4. Also, we deleted the word “complicated” from Table 4.

Line 205: “single-based PCR” what author mean by single-based? PCR for one target gene? Please speficy.

Response: We have added the explanation: single-based PCR (PCR for one target gene).

Line 207: “Clearly detected” – what authors mean by clearly detected?

Response: We agree this sentence was ambiguous. We intended to express the frustration of, for example having a suspicious culture but cannot test the infection immediately if we stick to conventional methods. We revised the sentence as follows: However, as these tests are time consuming, it is not realistic to perform the tests every time we find suspicious cultures.

Line 208: “We herein report a new rapid detection method for mycoplasma” it is better to use test than method. It is not completely new method. Add the inf. About A. laidlawii and U. urealyticum.

Response: As per your suggestion, we revised the sentence: We herein report a rapid detection test for class Mollicutes bacterial species, which can be easily performed in the laboratory.

Line 218-219: “examination showed” is repeated.

Response: It was repeated because conditions shown by two examinations were described in parallel. However, as per your suggestion, we changed one of the “showed” to “demonstrated”.

Line 247: It should be changed from “Mycoplasma strains from seven species” into seven strain of Mycoplasma spp.

Response: We revised the sentence.

Line 264: Specify what microscope.

Response: We have added the information.

Line 276: In my opinion better refer to Table 1.

Response: Thank you for your suggestion. We referred to Table 1 in the revised manuscript.

Line 281: Change “six DNA sources” into DNA isolates from six… and put the names of bacteria, virus and fungus in a correct order, because now it is mixed.

Response: We changed “DNA sources” to “synthetic DNA”. We changed the order of the names as you suggested.

Fig. 5. FAM and HEX are nor strictly detection wavelenghts but canals.

Response: We changed Fig. 5 with “Detection Channels: FAM (510 nm) and HEX (580 nm)”.

Line 291-292: In my opinion the information “according to the 17th revision of the Japanese Pharmacopoeia” is unnecessary in the conclusion. It is better to put the species examined with the test.

Response: We changed the sentences in the conclusion as you suggested.

Thank you so much for your helpful comments.

Reviewer 3 Report

Contamination of cell cultures by mycoplasmas (i.e. trivial name including genus Mycoplasma, Acholeplasma and Ureaplasma) is an important problem in cell-based transplantation. In order to avoid infection after transplantation, mycoplasma contaminations have to be detected before transplantation. Testing methods such as mycoplasma culture or DNA staining are time consuming and/or require well-trained technicians. The authors proposed to use the "myco finder PCR" for this purpose.

The main objectives of this paper were to test the sensitivity of the myco finder PCR with cell-cultures artificially contaminated with mycoplasmas, specificity using DNA and finally to test several cell lines with the myco finder PCR.

The authors looked for a rapid and simple procedure to test mycoplasma contaminations (lines 59-60). I understand that culture tests, DNA staining and nested PCR are time consuming or require well trained technicians, but how the Myco finder PCR is easier and faster that other PCR such as qPCR?

In this paper the myco finder PCR kit is described as a "new" PCR test. This is confusing because, the validation data for this PCR date of 2016, this is not a new PCR as it is commercially available for several years.

The main drawback of this paper is the absence of a comparison between the myco finder PCR versus other "reference" PCR or test on the same samples for the sensitivity and specificity tests. I think that a comparison on the same samples between myco finder PCR and qPCR as it was done for testing infected cell lines will be of great interest. Moreover, data indicating the advantages (time, procedure…) of the myco finder PCR compared to others PCR are incomplete. In Table 4, the data are without bibliographic references and incompletes (for instance, data regarding the time for "reference" PCR or qPCR are not described).

In the materials and methods section, 4 cell lines were used (lines 236-237), but only one is used for the sensitivity test, why only the iPS-RPE cell line is used ? How the authors confirmed that this cell line is mycoplasma-free before adding the mycoplasmas ? This is an important point to clarify. Moreover the description of the infected cell treatment for the myco finder PCR is not described as well as the PCR itself.

Why the specificity of the Myco finder PCR has been verified ? Such data have been already published on the website of Nissui pharmaceuticals (https://www.nissui-pharm.co.jp/english/pdf/products/global/Pharmacopoeia_Validation.pdf).

The SEM is not described in the materials and methods section. The cells line 46a (from monkey) is not described in materials and methods, only in the discussion. Lines 160-161: the authors stated that infected cells had less microvilli and were smaller, do the authors have any data of the average size of infected vs non-infected cells to support this observation ? How many cells have been observed for the microvilli and cells size ?

Lines 193-194: " …this PCR test was additionally conducted on the day of transplantation [2]": I did not find any reference of this PCR in cited reference "HLA-Matched Allogeneic iPS Cells-Derived RPE Transplantation for Macular Degeneration" ?

Lines 80, 99, 100, 249, 261 and Table 2: replace "M. arginine" by "M. arginini"

Author Response

Reviewer #3.

Contamination of cell cultures by mycoplasmas (i.e. trivial name including genus MycoplasmaAcholeplasma and Ureaplasma) is an important problem in cell-based transplantation. In order to avoid infection after transplantation, mycoplasma contaminations have to be detected before transplantation. Testing methods such as mycoplasma culture or DNA staining are time consuming and/or require well-trained technicians. The authors proposed to use the "myco finder PCR" for this purpose. The main objectives of this paper were to test the sensitivity of the myco finder PCR with cell-cultures artificially contaminated with mycoplasmas, specificity using DNA and finally to test several cell lines with the myco finder PCR.

Response: Thank you for your comments. We really appreciate. Yes, our objectives in this paper are as your comments above.

The authors looked for a rapid and simple procedure to test mycoplasma contaminations (lines 59-60). I understand that culture tests, DNA staining and nested PCR are time consuming or require well trained technicians, but how the Myco finder PCR is easier and faster that other PCR such as qPCR?

Response: Thank you for your comments. Compared with qPCR, Myco finder PCR is easier and faster. Please see new Table 4.

In this paper the myco finder PCR kit is described as a "new" PCR test. This is confusing because, the validation data for this PCR date of 2016, this is not a new PCR as it is commercially available for several years.

Response: As per your comments, we deleted the "new" throughout the manuscript.

The main drawback of this paper is the absence of a comparison between the myco finder PCR versus other "reference" PCR or test on the same samples for the sensitivity and specificity tests. I think that a comparison on the same samples between myco finder PCR and qPCR as it was done for testing infected cell lines will be of great interest. Moreover, data indicating the advantages (time, procedure…) of the myco finder PCR compared to others PCR are incomplete. In Table 4, the data are without bibliographic references and incompletes (for instance, data regarding the time for "reference" PCR or qPCR are not described).

Response: Actually, we have done the mycoplasma quantitative PCR (qPCR) test using retinal samples. We previously reported mycoplasma quantitative PCR (qPCR) test (reference No. 15). As per your suggestions, we added mycoplasma qPCR in Table 4 for comparison with myco finder PCR.

In the materials and methods section, 4 cell lines were used (lines 236-237), but only one is used for the sensitivity test, why only the iPS-RPE cell line is used ? How the authors confirmed that this cell line is mycoplasma-free before adding the mycoplasmas ? This is an important point to clarify. Moreover the description of the infected cell treatment for the myco finder PCR is not described as well as the PCR itself.

Response: The validation of one type of mycoplasma required at least 1.6 x 10E7 RPE cells in this study. So, we chose a RPE line used in clinical trials and used them throughout the study. Before the validation test in the present study, we confirmed "negative" by mycoplasma quantitative PCR. We added this information in Method section 4.4. We also described the pretreatment of infected cells for Myco finder PCR in Method section 4.1.

Why the specificity of the Myco finder PCR has been verified ? Such data have been already published on the website of Nissui pharmaceuticals (https://www.nissui-pharm.co.jp/english/pdf/products/global/Pharmacopoeia_Validation.pdf).

Response: The reason for verifying the specificity of Myco finder PCR is for our institutional validation. We wanted to confirm the specificity and the sensitivity by our own criteria. However, because we thought that it is also important, we introduced the specificity data provided by Nissui Pharmaceutical in Method section 4.4.

The SEM is not described in the materials and methods section. The cells line 46a (from monkey) is not described in materials and methods, only in the discussion.

Lines 160-161: the authors stated that infected cells had less microvilli and were smaller, do the authors have any data of the average size of infected vs non-infected cells to support this observation ? How many cells have been observed for the microvilli and cells size ?

Response: The SEM is now described in the materials and methods section 4.5 in the revised manuscript. Because RPE cells are often larger than the field of view in SEM (representative image in Fig 4C upper panel), it is difficult to specify the accurate cell size. So instead of measuring the average size of infected and non-infected cells, we counted the cell number of infected and non-infected cells in one field of view (X3000). Six views each for infected and non-infected were counted. Statistical test was also performed. Please see new Fig. 4C, D. We observed 6 SEM images each for infected and non-infected cells (representative image of each cell condition is shown in Fig 4C), and we failed to find any microvilli on the surface of infected RPE cells. We revised the manuscript with the information above.

Lines 193-194: " …this PCR test was additionally conducted on the day of transplantation [2]": I did not find any reference of this PCR in cited reference "HLA-Matched Allogeneic iPS Cells-Derived RPE Transplantation for Macular Degeneration" ?

Response: It is described on page 4 and Table S8 (page 25) of the Supplemental file of Reference No. 2 (HLA-Matched Allogeneic iPS Cells- Derived RPE Transplantation for Macular Degeneration). Information on the web site is below: https://www.mdpi.com/2077-0383/9/7/2217/s1.

Lines 80, 99, 100, 249, 261 and Table 2: replace "M. arginine" by "M. arginini" 

Response: We corrected "M. arginine" to "M. arginini ". Thank you so much for your helpful comments.

Round 2

Reviewer 3 Report

Review IJMS1459658-V2

I would thank the authors for answering the questions that have been raised.

Please checked for Figure 4 (C) vs lines 188-189: upper is mycoplasmas – and lower mycoplasmas +.

This manuscript is a resubmission of an earlier submission. The following is a list of the peer review reports and author responses from that submission.

Round 1

Reviewer 1 Report

The authors present a novel PCR method for detection of mycoplasma contamination in transplantanted eye cells.  The presented method seems to be more simple and sensitive compared to the conventional test. The manuscript is generally well written and the study was well designed. However, a few small comments should improve the quality of the manuscript, as follows:
- 'spp' in 'Mycoplasma spp.' should be written with no italic (e.g. page 2, 5)
Matherial and Methods:
- page 7: the authors should give more detailed information about the methods used for the CFU and GC determination
- the authors should answer if the samples used to validate the presented method were tested using a different molecular method? 

Author Response

The authors present a novel PCR method for detection of mycoplasma contamination in transplantanted eye cells. The presented method seems to be more simple and sensitive compared to the conventional test. The manuscript is generally well written and the study was well designed. However, a few small comments should improve the quality of the manuscript, as follows:
- 'spp' in 'Mycoplasma spp.' should be written with no italic (e.g. page 2, 5)
     Response:  We changed them without italic.

Matherial and Methods:
- page 7: the authors should give more detailed information about the methods used for the CFU and GC determination

     Response:  As per your suggestion, we have described detailed methods used for the CFU and GC determination in method section (new section 4.3) in the revised manuscript.

- the authors should answer if the samples used to validate the presented method were tested using a different molecular method? 

     Response:  We agree with your comments. In fact, we also examined the samples using quantitative real-time PCR methods as well. We present the data (section 2.3) and describe the methods (section 4.4) in the revised manuscript.

Thank you for your helpful comments.

Reviewer 2 Report

Review of the manuscript suggests that this is the preliminary phase of the development of a new PCR test, for which not all the required information, such as the newly developed primers, is provided.

The text of the manuscript is full of inaccuracies, and, despite not being an expert, requires revision of the English language. The manuscript would need a thorough revision and the addition of more information.

Regarding the test, it would possibly require the analysis of DNA from other bacteria to be sure of the specificity, as it is based on the 16s RNA gene. In addition, it is necessary to know how many samples from cell lines were used in the study to check the new PCR system, as this is essential to know the robustness of the results. Moreover, the results should be compared with the system used so far, but analysing a sufficient number of samples. 

Some arbitrariness or lack of information is detected:

"In addition, the specificity of this PCR test was also high". 
"This new test is sufficiently simple, fast, and sensitive that we will validate it in our next RPE clinical trial as a quality control".

Only two references are used in the introduction. There are no references for some paragraph. It should be included to justify the data.

For example, the information shown in table 3 regarding the tests available to detect contaminants are not consistent with the information available regarding the use of PCR tests to detect mycoplasmas. It does not require almost 7 days for this, or the mandatory use of a Nested-PCR. Moreover, it is necessary to know how many samples from cell lines were used in the study to check the new PCR system.

Author Response

Review of the manuscript suggests that this is the preliminary phase of the development of a new PCR test, for which not all the required information, such as the newly developed primers, is provided. The text of the manuscript is full of inaccuracies, and, despite not being an expert, requires revision of the English language. The manuscript would need a thorough revision and the addition of more information.

     Response: Thank you for your comments. The primers used for the PCR was not provided with the Myco Finder PCR kit because of the patent issue. We have had a native speaker with expertise in molecular biology and infection revise the manuscript.

Regarding the test, it would possibly require the analysis of DNA from other bacteria to be sure of the specificity, as it is based on the 16s RNA gene.

     Response: As per your concerns, we examined additional 16 bacterial DNA with this PCR to be sure of the specificity. Please see new Table 3.

In addition, it is necessary to know how many samples from cell lines were used in the study to check the new PCR system, as this is essential to know the robustness of the results. Moreover, the results should be compared with the system used so far, but analysing a sufficient number of samples. 

     Response: We agree with your comments. In the revised manuscript, we added the information how many samples from each cell line were used to check the new PCR system (lines 143-146).

Some arbitrariness or lack of information is detected:

"In addition, the specificity of this PCR test was also high". 

     Response: As described above, the specificity was confirmed by the analysis of 16 additional bacterial DNA (new Table 3). We also have revised the sentences in the abstract.

"This new test is sufficiently simple, fast, and sensitive that we will validate it in our next RPE clinical trial as a quality control".

     Response: We omitted the sentence from the abstract.

Only two references are used in the introduction. There are no references for some paragraph. It should be included to justify the data.

     Response: As per your concerns, we have added several references including a new one in the introduction.

For example, the information shown in table 3 regarding the tests available to detect contaminants are not consistent with the information available regarding the use of PCR tests to detect mycoplasmas. It does not require almost 7 days for this, or the mandatory use of a Nested-PCR.

     Response: We apologize for the lack of information. We clarified this in the table footnote as follows: Conventional tests indicate our previous conventional test for detecting mycoplasma species.

Moreover, it is necessary to know how many samples from cell lines were used in the study to check the new PCR system.

     Response: As described above, we added the information how many samples from each cell line were used in the study to check the new PCR.

Thank you for your helpful comments.

Reviewer 3 Report

why they don't detect all type of mycoplasm as well as ureoplasm or mycoplasma hominis since  they test a culture of retinal cells?

which kind of mycroscope they used in picture 3?

the scale bar for mycoplasma it's to high to identify mycoplasma, we suggest to use a SEM for negative or positive control, I suggest to compare the mycofinder PCR test with Mycoallert (more used in cell culture laboratory).

I suggest to utilize agar for positive control.

In consideration of this I consider the work uncomplete.

Author Response

why they don't detect all type of mycoplasm as well as ureoplasm or mycoplasma hominis since they test a culture of retinal cells?

     Response: Regarding Ureaplasma and mycoplasma hominis, Ureaplasma urealyticum (ATCC 27618) and mycoplasma hominis (NBRC 14850) were tested by this PCR assay and were confirmed to be detectable. We have added the results in the revised manuscript (lines 87-89).

which kind of mycroscope they used in picture 3?

     Response: We used an inverted microscope (IX71, OLYMPUS, Tokyo, Japan). We have added this in Fig. 3 legend and method section.

the scale bar for mycoplasma it's to high to identify mycoplasma, we suggest to use a SEM for negative or positive control, I suggest to compare the mycofinder PCR test with Mycoallert (more used in cell culture laboratory). I suggest to utilize agar for positive control. In consideration of this I consider the work uncomplete.

     Response: Thank you for the comments. Regarding the scale bar, it is not necessary to observe by scanning electron microscope (SEM) because visualization of mycoplasma in cell culture is not the purpose of the study. Instead, we thought detection of mycoplasma in cultured cells whose changes after the infection are not observed with an optical microscope is necessary. We clarified this in the Results section (lines 134-137) as follows: When the infected B95-8 cells were observed under an inverted microscope, as shown in Fig. 3B, they were indistinguishable from uninfected B cells. By quantitative real-time PCR (qPCR), the infection of mycoplasma spp. in these B95-8 cells was calculated as 4.4 × 10E6 copies/μL.

     For the same reason, we don't think positive control with agar is necessary. Regarding MycoAlert, it is not recognized by the Japanese Pharmacopoeia 17th revision, so we think the comparison of Myco Finder with MycoAlert may not be meaningful considering the purpose of this study that is to build a rapid and simple PCR system for the detection of mycoplasma in transplants for clinical use.

Thank you for your helpful comments.

Round 2

Reviewer 2 Report

After revising the new version of the manuscript, it has been substantially improved, incorporating much of the required information. In this sense, before the final acceptance of the work, I recommend the incorporation in the table 1 of the information regarding the detection of Ureaplasma urealyticum (ATCC 27618) and M. hominis, as well as the elaboration of a table similar to those described (table 2) for the rest of the mycoplasma species, in order to avoid confusion to the readers and to facilitate the reading of the work.

Reviewer 3 Report

raw 89: needs data on Ureaplasma and Micoplasma Hominis to sustain that PCR assay is valid. 

raw 135: inverted microscope it's wrong machinery to identify mycoplasma it's resolution power is to late, so you must repeat experiment using another microscope like confocal (for seeing presence of eggfries colony) or SEM (for seeing single units of mycoplasma) the actual figures are not reliable. You must use a correct control.

raw 163: If you test PCR assay for Ureoplasma and M. Hominis why you sign just seven mycoplasma species?